# Analyzing technology acceptance and perception of privacy in ambient assisted living for using sensor-based technologies

Wiktoria Wilkowska[1]*, Julia Offermann[1], Susanna Spinsante[2], Angelica Poli[2‡], Martina Ziefle[1‡]

1 Human-Computer Interaction Center, RWTH Aachen University, Aachen, Germany, 2 Department of Information Engineering, Università Politecnica delle Marche, Ancona, Italy

☯ These authors contributed equally to this work.
‡ AP and MZ also contributed equally to this work.
* wilkowska@comm.rwth-aachen.de

**Data Availability Statement:** All relevant data are within the manuscript and its Supporting Information file (see S1 Data).

## Abstract

People increasingly use various technologies that enable them to ease their everyday lives in different areas. Not only wearable devices are gaining ground, but also sensor-based ambient devices and systems are increasingly perceived as beneficial in supporting users. Especially older and/or frail persons can benefit from the so-called lifelogging technologies assisting the users in different activities and supporting their mobility and autonomy. This paper empirically investigates users' technology acceptance and privacy perceptions related to sensor-based applications implemented in private environments (i.e., passive infrared sensors for presence detection, humidity and temperature sensors for ambient monitoring, magnetic sensors for user-furniture interaction). For this purpose, we designed an online survey entitled "Acceptance and privacy perceptions of sensor-based lifelogging technologies" and collected data from $N = 312$ German adults. In terms of user acceptance, statistical analyses revealed that participants strongly agree on the benefits of such sensor-based ambient technologies, also perceiving these as useful and easy to use. Nevertheless, their intention to use the sensor-based applications was still rather limited. The evaluation of privacy perceptions showed that participants highly value their privacy and hence require a high degree of protection for their personal data. The potential users assessed the collection of data especially in the most intimate spaces of domestic environments, such as bathrooms and bedrooms, as critical. On the other hand, participants were also willing to provide complete data transparency in case of an acute risk to their health. Our results suggest that users' perceptions of personal privacy largely affect the acceptance and successful adoption of sensor-based lifelogging in home environments.

## Introduction

The current development in information and communication technologies enables a variety of applications in many different contexts. Assisting technologies such as wearable devices as well

**Funding:** This work is part of the PAAL-project (Privacy-Aware and Acceptable Lifelogging services for older and frail people). In particular, the support of the Joint Programme Initiative More Years, Better Lives (award number: PAAL_JTC2017), the German Federal Ministry of Education and Research (grant no: 16SV7955), and the Italian Ministero dell'Universitá e della Ricerca, (CUP: I36G17000380001) is gratefully acknowledged. The funders had no role in study design, data collection and analysis, decision to publish, or preparation of the manuscript.

**Competing interests:** The authors have declared that no competing interests exist.

as lifelogging applications [1] capable of assisting users in their (smart) home environments or in Ambient Assisted Living (AAL) institutions allow for meaningful support in different areas of life [2, 3]. Especially within the context of healthcare and social care, lifelogging technology opens up far-reaching opportunities for preventive medicine, diagnostics, therapeutical/rehabilitation measures and allows controlling healthcare cost. Particularly for older users, lifelogging technologies support autonomous living for longer time periods.

Modern technologies reach not only young and tech-savvy people who know how to use the almost daily newly emerging applications in their private and professional lives, but also those who did not directly grow up with the technology, having rather acquired its use at some later point in life. Especially older and/or frail persons can significantly benefit from the potential offered by assistive technologies in their private settings by increasing their health-related safety and maintaining their physical well-being and mobility.

Besides general applications that allow monitoring of physical activities, daily habits [4], vital parameters [5], or the detection of emergencies (e.g., [6]), older and/or frail persons can also use age-specific or illness-specialized functions. For instance, sensor-based technologies enable identifying changes in behavior, movement and sleeping patterns, or walking speed [7]; they also provide potential indicators for the recognition of diseases, such as dementia or Parkinson's disease (e.g., [8, 9]). Hence, analyses of such data enable an early identification of deviations from the normal behavior, health changes, or the detection of possible abnormalities in bodily functions as well as provide security for the users and their important others in the long run.

Despite the availability of different assistive applications, however, the main questions arising in this context are whether individuals are prepared for adopting such assistive technologies, to what extent they intend to use them, and where the possible barriers may lie. On the other hand, the lifelogging of (health-related) data from wearable or ambient technologies significantly interferes with the users' privacy and experience of autonomy. Research has shown that invasion of privacy poses one of the main concerns in the adoption of technologies (e.g., for voice-based digital assistants: [10]) as the assistive technologies capture a great amount of private information that people may not want to share with others [11].

Considering the fast pace of emerging assistive applications or systems, the current stage of research is insufficient when it comes to the ethical, legal, and social implications of using sensor-based technologies in home environments. There is a lack of a context-sensitive analysis of user acceptance and an empirically based exploration of the users' requirements for the recording and managing of (health-related) user data referring to their privacy. In this paper, we therefore address the current gap in research and empirically investigate the users' attitudes towards acceptance and privacy in the context of sensor-based technologies in private environments.

## Theoretical background

In the following, we firstly describe the technology under study in more detail, and we highlight the theoretical background that provides the basic framework for the empirical investigation of user acceptance and perceptions of privacy for the adoption of sensor-based lifelogging technologies from the user's perspective.

### Lifelogging technologies

As we focus in this study on lifelogging technologies, in the first step, we briefly outline the definition of the term. Selke [1] described "lifelogging" as different types of digital self-tracking and recording of everyday life and a phenomenon that can be placed between innovative

technologies and cultural transformation. Lifelogging refers to diverse types of self-tracking, ranging from health prevention and the monitoring of bodily functions over one's location and motion to the measurement of productivity at work and in private life. Thus, the technology can capture and record certain parts of a human life in real-time, saving them for later use and analyses.

With this in mind, lifelogging technologies have a great potential to manage many of the concerns raised by population aging. The applications can be used to prevent disease, provide personalized healthcare, and give support to formal and informal caregivers [12]. There are different types of technologies, which can be applied for the mentioned purposes: Most common are wearable devices, such as smart watches and wristbands, often used for the monitoring of physical activities, location determination, and general self-awareness. This trend of using lifelogging for self-improvement is predominantly associated with younger and physically fit adults [13, 14] and it gains particular interest when it comes to the topics of health prevention and fitness. However, lifelogging systems can also provide great support for older and/ or frail people by integrating them into the everyday life of persons in need of help. This means that lifelogging applications can not only be worn on the body, but they can be also integrated into a domestic space in order to capture useful data; in this context, the technology is then classified as AAL technology [15], shifting the focus to the function of assistance rather than self-improvement. Information gained in this way can be monitored by caring relatives and/or nursing personnel, and the data can be further analyzed by professionals to identify irregularities or emergencies, or even to predict changes in behavior or health status [16].

Generally, the process of lifelogging is considered a passive and non-intrusive activity which can be carried out by various types of lifelogging systems [13], ranging from simple trackers over (wearable) cameras and microphones to ambient sensors integrated into living environments. Accordingly, the recorded data can consist of multimedia data like video and audio recordings as well as sensor data [17].

## Ambient sensor-based technologies used in the study

As mentioned above, despite being mostly referred to the domain of wearable sensors and devices, lifelogging may also rely on sensing infrastructures or sensor systems displaced in living environments. Setting up an ambient sensor-based lifelogging solution requires the installation of hardware devices in the living environment and their connection to a gateway that acts as a data aggregator and relay to a remote server. Typically, on the remote server, the received data is organized and presented to the user by a dashboard, from which several functionalities of reporting, analytics, and visualization are accessible.

Focusing on the selection of ambient sensors and their installation, a basic lifelogging setup, like the one considered in this study, may be obtained by using a small set of simple sensors, such as Passive InfraRed sensors (known as PIR)—to capture motion and presence in a given environment or room—, relative humidity and temperature sensors (H&T), and magnetic sensors (MAG) to detect doors and windows opening and closing but also user's interaction with drawers or furniture, depending on the specific installation location. Both PIR and MAG sensors may be classified as binary sensors, as their output can reflect two possible states only, such as presence/no presence or motion/no motion, and open/close, respectively. H&T sensors, on the contrary, provide numerical information that can be used not only to monitor indoor comfort, but also to detect specific events in the living environments, correlated to the user's habits. All these types of sensors are easily found in the market and sold by several different vendors, with prices starting from a few Euros to several tens Euros per device, depending on the brand chosen, the desired accuracy and precision of the sensing capability, the

availability of additional services supported by the vendor, usually through a mobile app and a dedicated server for which the user has to pay a monthly or annual access fee. For a quick and minimally intrusive installation, nowadays almost all the sensors are battery-powered and enabled with a Bluetooth Low Energy (BLE) or ZigBee wireless connection to a specific device, named gateway (or concentrator), in charge of collecting the data generated by many sensors and relay it to the remote server, usually over a WiFi link.

In our study, the PIR sensor was located inside the bathroom, on the top of the mirror, in order to identify the events corresponding to the user entering and exiting the bathroom. According to the specifications provided by the manufacturer, the detectable distance amounts to 7 m, with a 170˚ field of view. The chosen H&T sensor can detect a 0.3˚C temperature variation in the range [0, 60] ˚C, and a 3% variation of the relative humidity in the range [0, 99] %. The sensor was located inside the kitchen to detect moments of the day related to food preparation (i.e., cooking events). MAG sensors have been applied to drawers and furniture doors to collect data able to describe the way people interact with the living environment: specifically, one MAG sensor was applied onto the fridge door, another one onto the drawer containing silverware. All the sensors were bought together, in a kit including the gateway as well, for around 80 Euros. An associated free-to-download Android mobile app is available, to set up and configure the sensors, and to visualize the list of collected data, but no additional services or remote dashboard are provided.

## Technology acceptance of sensor-based lifelogging technologies

In the past, many scientific studies have shown that the adoption of a relevant technology is not solely determined by its technical merit. User acceptance is almost equally decisive for the successful establishment. However, the technology acceptance is a complex construct, and its nature may differ depending on the context under study.

In many years of research on technology acceptance meaningful models have been developed. To date, the Technology Acceptance Model (TAM, [18]) and the Unified Theory of Acceptance and Use of Technology (UTAUT; [19, 20]) have become established in this area. In this paper we focus on the TAM as this model has built the theoretical background for the current study. TAM is founded upon the hypothesis that the technology's acceptance and use can be explained in terms of the user's internal beliefs, attitudes, and intentions [21]. The model assumes that an individual's behavioral intention to use a technology or an information system is determined by the perceived ease of use (PEU) and perceived usefulness (PU). Davis [18] defined PEU as "the extent to which a person believes that using the system would be free of effort" (p. 320), claiming that an application perceived as easy to use is more likely to be accepted by the user. Further, the author defined PU as "the extent to which a person believes that using the system would enhance his or her job performance" ([18], p. 320) and assumes that technology or application that is perceived as highly useful leads to a positive use-performance relationship. With this in mind, technology considered useful is the one of which users believe that it helps them to perform the job, planned function, or action better. The model theorizes furthermore that PU is connected with PEU, because the easier the technology's use is, the more useful it can be perceived [22]. In addition, TAM claims that external variables indirectly influence technology use through PEU and PU, which affect the user's attitude (A) towards, and the behavioral intention to use (ItU), the technology. The theoretical model is schematically presented in Fig 1.

Even though the mentioned theoretical models have found very wide deployment in the field of information and communication technology, mainly addressing the job-related context of using, they do not satisfactorily evaluate all kinds of the currently available technologies.

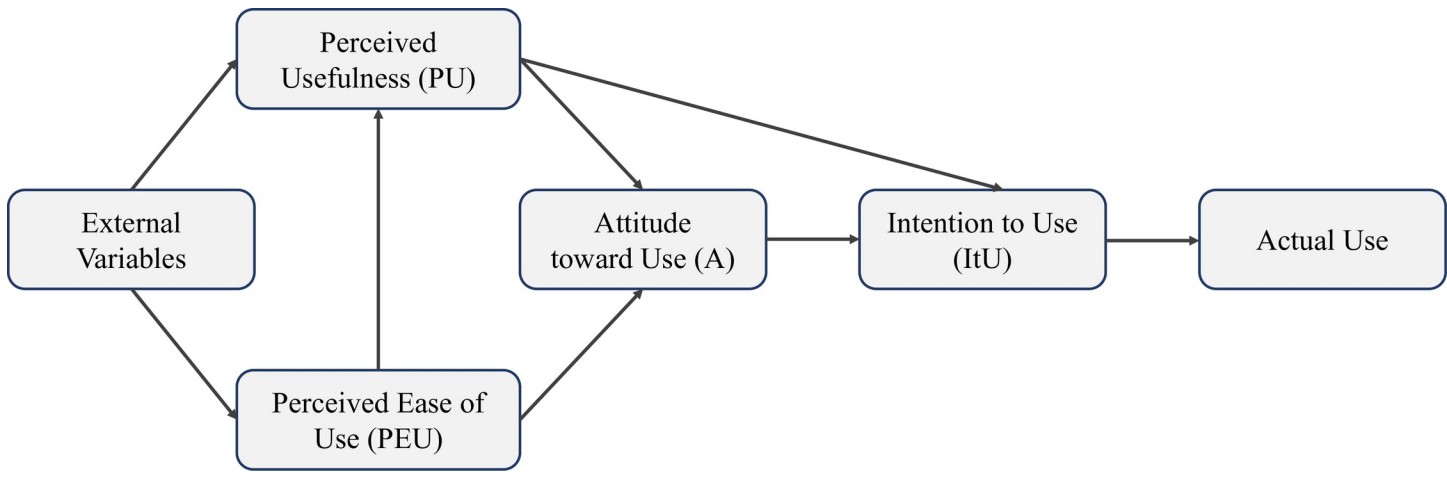

**Fig 1. Technology acceptance model** [18].

Especially in the area of the thriving lifelogging technologies, more specific opinions of the potential users are needed for the evaluation of the intention to use, and the real use, of the wearable or ambient applications. Previous research showed that while the perception of benefits can essentially encourage the utilization of a targeted technology, the awareness of barriers can significantly diminish the intention to use it (e.g., [23, 24]). Lifelogging applications which have the potential to assist the users—especially the older and/or frail ones—in the everyday life are mostly appreciated for the ability to support autonomy (e.g., [25]), the improvement of the sense of safety and security (e.g., [26, 27]), and easing the burden of caring family members [23, 28, 29]. Continuing within the context of aging in place, use of lifelogging applications can also be associated with significant concerns. Invasion of individual privacy, stigma, and feelings of isolation are important barriers in terms of using assistive lifelogging in the daily routine (e.g., [12, 30, 31]). As to data security, especially the perceived loss of control over the sensitive data, an unauthorized access to, or misuse of, that data by third parties represent great barriers to the use of lifelogging applications [11, 32].

Besides the perceived motives and barriers in the process of the adoption of assistive technologies, the properties of the technology itself as well as the characteristics of the users can have a decisive impact on the acceptance. Himmel and Ziefle [33] provided evidence that cameras were less desired as assistive lifelogging technology in comparison to motion sensors or microphones. The preferences for certain modes of technology were also validated in a study of emergency detection that showed wearable devices, such as smart watches or emergency buttons, being clearly more accepted by the users than cameras [34]. Notwithstanding, a study of Gelonch et al. [35] showed that older adults diagnosed with mild cognitive impairment can become competent users of lifelogging wearable cameras with a good level of acceptance. Hence, the adoption and actual use of different assistive applications clearly depend on the specific functionality and the circumstances of use, on the one side, and the characteristics of the receivers, on the other side.

### Requirements of personal privacy

Perceptions of individual privacy pose a specific concern in the adoption of lifelogging technologies. As these technologies enable the user to capture and publish his/her personal data with ease, they also pose a major challenge in terms of maintaining the user's privacy.

The right to privacy is one of the fundamental human rights in a modern society and in the European Union it is protected by the General Data Protection Regulation (GDPR). According to Goddard [36], the GDPR has six core privacy principles that concern the general data protection: (1) fairness and lawfulness, (2) purpose limitation, (3) data minimization, (4) accuracy, (5) storage limitation, and (6) integrity and confidentiality. However, data protection by design and default is at the core of the GDPR and it is supported by transparency (full information is provided to individuals) and accountability (organizations take demonstrable responsibility using personal data). Mihailidis and Colonna [12] proposed the application of Privacy by Design (PbD) as one way to manage privacy concerns raised by lifelogging technologies. This approach involves the transformation of legal rules, namely those that pertain to privacy and data protection, into information systems and allows for incorporating privacy into all stages of a lifelogging system's development, considering the contextual understanding and the specific design elements of these kinds of systems.

Depending on the type of the lifelogging application, differently large amounts of user data are collected and stored. Research has shown that the expectation of privacy for the lifelogger can differ, depending on the context of use and the type of the used technology. As already mentioned above, video-based lifelogging poses a major hurdle to the acceptance and use of a technology [33, 34]. In this field, Padilla-López et al. [37] proposed a visual privacy by context approach that strives for a trade-off between the privacy preservation and the intelligibility of images being acquired. With this approach, the privacy protection can be constituted by different visualization modes, which conceal the sensitive information to a variable extent, providing thus context-dependent levels of protection of the lifelogger [38]. According to different studies on the acceptance of assistive technologies, privacy concerns, especially those regarding the sense of surveillance and misuse of data access, are confirmed to be a barrier (e.g., [39]). The trade-off between the benefits provided by assistive technology (e.g., increased security and preservation of autonomy) and the loss of privacy is of major interest in acceptance research as it provides valuable insights into the (potential) users' decision processes towards the intention to use a particular technology. Research of Jaschinski et al. [40] revealed for instance that for older users, who are concerned about their privacy when using assistive technology, the benefits of increased safety and staying in their own home are significantly more relevant, and thus outweigh the feeling of the privacy loss. At this point, it becomes even clearer how sensible legal regulations are according to the PbD approach.

In the context of the present study, we investigate the perceptions of privacy with respect to ambient sensor-based lifelogging technologies considering not only different installation sites of the technology but also the different circumstances of the users. We additionally review how, in this context, individual privacy is related to the acceptance of this technology.

## Objectives and aim of the study

On the basis of the presented considerations, previous research on the acceptance of assisting lifelogging technologies predominantly focused on video- and audio-based systems, and comparably few studies investigated the adoption of sensor-based technologies. In this study, we empirically investigate the users' attitudes towards acceptance and their perceptions of personal privacy in the context of sensor-based lifelogging in private environments. The underlying research questions of the study are the following:

RQ1: How high is the general acceptance of sensor-based technology in home environments: Do the potential users intend to use it? How do they perceive the technology's usefulness and ease of use?

RQ2: Are potential users more likely to be guided by perceptions of the benefits rather than the drawbacks that the technology brings?

RQ3: Which role does privacy play in the adoption of sensor-based ambient technologies and do the privacy perceptions change depending on the user's health condition?

RQ4: Does the protection of individual privacy have a significant impact on the acceptance and the willingness to use sensor-based technology?

## Methods

In this section, we describe the methodological approach of our empirical study, introducing the structure of the online survey, the used study design, description of the sample, and the applied data analysis procedures.

### Data collection

The study involving human participants was reviewed and approved by The Ethics Committee (Division 7.3) "Empirical Human Sciences" at the Faculty of Humanities at RWTH Aachen University (ID: 2021_003_FB7_RWTH Aachen). Prior to the survey beginning, participants were informed about the study's aim and content and provided their written consent to the scientific handling of their indicated data and to the participation in the study. Data collection was realized by means of a standardized online survey, which was performed between May and June 2021, using the professional platform Qualtrics XM.

The online survey was divided into three parts. The first part referred to the participants' demographic characteristics, i.e., age, gender, educational level, civil status, and housing situation. After this, participants described their own health condition [i.e., a general health status (ranging from 'very good' to 'very poor')], chronic disease ('yes'/'no'; if 'yes': the need for support in everyday life), and their experiences of caregiving (professional vs. private experience in care).

The second part of the survey provided general information about sensor-based lifelogging technologies and systems. Here, the wide application spectrum of sensor-based lifelogging solutions was presented, pointing out both the common wearable and home-integrated sensors. As the focus of the study was to evaluate the acceptance of ambient sensors that assist (older/frail) inhabitants in their everyday life, concise information—previously elaborated by an interdisciplinary collaboration between engineering and communication science experts—was presented to the participants using descriptions and illustrations as presented in Fig 2. Therefore, the technical feasibility and the simple operability (on/off-functions) were exposed on the one hand, and all the potentials resulting from the possible overviews of daily habits, activities, and behaviors, as well as long-term comparisons, and useful profiles of different users, on the other. After receiving all the general details, participants were instructed to empathize with the situation, in which the introduced sensors are installed within their own home environment. Subsequently, they assessed the potential benefits (e.g., "ability to live independently at home") and barriers (e.g., "concerns about false alarms") of such lifelogging technology in their domestic environments. In addition, the evaluation of technology acceptance was extended by questions referring to criteria postulated by TAM [18]: perceived ease of use, perceived usefulness, and intention to use (6 items; Cronbach's $\alpha$ = .76). All acceptance items were assessed on six-point Likert type scales ranging from "fully disagree" (= 1) to "fully agree" (= 6). Before starting with the next part, the survey evaluated the relevance of health in the perception of privacy when using sensor-based technology (the four items are presented in Fig 9; $\alpha$ = .80), using the same (dis)agreement scale as for the acceptance statements.

**Part I:**
**Characteristics of**
**Participants**

- Demographics (age, gender, education, civil status)
- Housing situation
- Health condition
- Previous experience with care

**Part II: Information about ambient sensors & the evaluation of user acceptance**

Passive infrared sensors

Humidity and temperature sensors



- Perception of benefits
- Perception of barriers
- Perceived ease of use
- Perceived usefulness
- Intention to use

Magnetic sensors

R a n d o m i z a t i o n

**Part III:**
**Perceptions of**
**privacy**

- General perceptions of privacy using the sensors
- Privacy issues connected to specific activities when using sensor-based systems at home
- Relevance of health in the perception of privacy

**Fig 2. Schematic structure of the online survey.**

Finally, the third part of the survey concentrated on the users' perceptions of privacy in the context of using sensor-based technology in home environments. First, participants rated their general perceptions of privacy in this context (e.g., "Privacy protection should be the first priority in sensor-based lifelogging.") using agreement scales as described above. Subsequently, the questionnaire collected opinions on privacy aspects connected to different specific activities, such as recordings of the stay in different premises, timing, frequency and duration of individual activities, as well as generating of (temporal) profiles. Participants used a slider to rate how critical—in their eyes—such recordings of the respective activities are (from 0 = "not critical at all" to 100 = "very critical"). Fig 8 summarizes all the activities.

To ensure a high comprehensibility of the technical explanations and enable unambiguous projections of the technology object(s) to be evaluated, the survey was thoroughly pretested prior to the launch of the study. The respondents participated on a voluntary basis in this survey. Only adults were allowed to participate in the survey after they consented to a statement regarding the data protection and data retention.

## Study design

The intention of this research was to examine the willingness to adopt sensor-based technologies in the private living environments. In the survey, we provide respondents with concrete information about the technology, and investigate to what extent such assistive technologies are accepted and permitted, or what conditions (e.g., privacy, data protection) need to be met in order to increase the acceptance among potential users.

Considering this, for the present study two main aspects are in focus: Firstly, the technology acceptance including the general perceptions of benefits (6 items; α = .90) and barriers (6 items; α = .83) associated with the use of sensor-based ambient technologies preferences as well as aspects like the perceived usefulness, ease of use, and the intention to use such sensors, applications, or systems; the later are described in detail in Table 1, while benefits and barriers are summarized in the Figs 4 and 5.

Secondly, the general perceptions of privacy as well as privacy in dependence on health pose another main aspect in the context of the sensor-based systems in this study; the used items are provided in Figs 7 and 9. To deepen the concept of privacy, the privacy assessments of individual activities when using ambient sensors complement the analyses: The respondents' evaluations of how critical they perceive the specific events/activities (see Fig 8) enable even more insights on privacy in the context of sensor-based lifelogging.

Both the mentioned aspects—technology acceptance and privacy perceptions—are assumed to represent the core premises of a successful adoption of the sensor-based technology, and are examined in this research in more detail. The study design is depicted in Fig 3.

## Sample description

After a quality inspection and data cleansing of our quantitative study, data from a total of N = 312 individuals were finally taken for the statistical analyses. The sample was rather young (M = 32.9 years; SD = 18.7) with participants ranging in their age between 18 and 91 years. There was a higher proportion of females (62.8%; n = 196) compared to males (36.9%; n = 115) in the sample, while one participant indicated a diverse gender (0.3%). On average, the participants were comparatively highly educated with 55.8% (n = 174) of them holding a university entrance qualification and 28.2% (n = 88) holding a university degree. Only 16.0% of the participants (n = 50) reported lower educational levels, such as secondary or elementary school certificates. As to civil status, most participants reported to live in a partnership or be married (56.4%; n = 176), one third (33.3%; n = 104) indicated to be single, and small parts of the sample were divorced (1.9%; n = 6) or widowed (4.2%; n = 13); the rest (4.2%, n = 13) did not indicate their civil status.

As the research focuses on the use of ambient sensor-based technologies, which, due to their multiple applications and support possibilities, can be beneficial especially for ill and/or elderly users, the health status of the participants was also of interest. The vast majority of the participants indicated a (very) good health condition (83.9%; n = 262). One fifth of all participants (20.2%, n = 63) reported to suffer from a chronic disease or have a physical disability or limitation. Accordingly, solely six respondents (1.9%) indicated to need assistance and support

**Table 1. Items used for technology acceptance [18] adapted to sensor-based lifelogging technology and the internal consistency (Cronbach's α) of the particular criteria.**

| | Items | Reliability |
|---|---|---|
| Perceived usefulness (PU) | "It is useful to get an overview of the own activities with the help of sensor-based technologies." <br> "With the help of sensor-based technologies, parts of life can be optimized." | α = .63 |
| Perceived ease of use (PEU) | "The use of sensor-based technologies seems very simple to me." <br> "The use of sensor-based technologies is too complicated." [recoded] | α = .76 |
| Intention to use (ItU) | "I would not use a sensor-based lifelogging system for support in my home." [recoded] <br> "I intend to use a sensor-based lifelogging system to assist me in my home in the future." | α = .75 |
| | Overall | α = .75 |

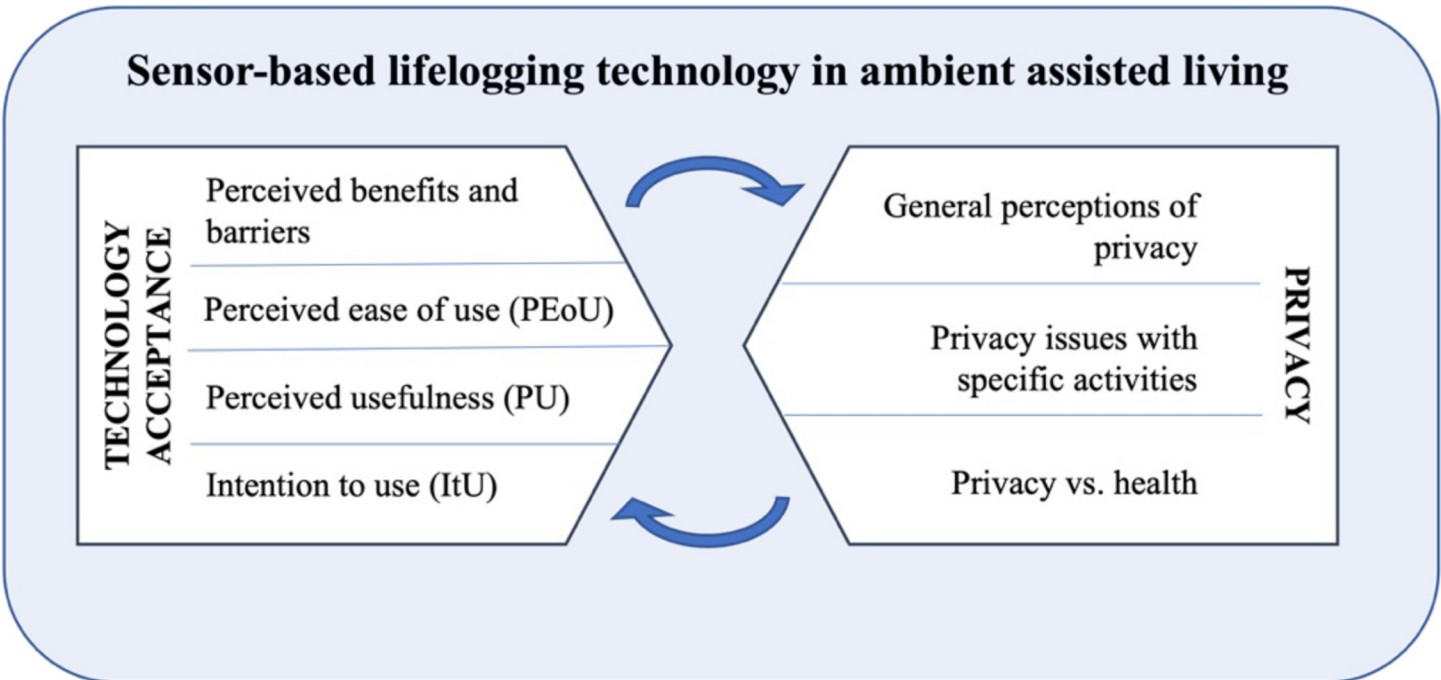

**Fig 3. Study design.**

in their everyday life. Asked for experiences in care, 17.3% of the sample (n = 54) reported a professional experience in care, while 24% (n = 75) of them indicated an active private experience and 31.7% (n = 99) a passive private experience in caring for a person/family member in need of care. Regarding previous experiences in using lifelogging applications, respondents reported to use lifelogging for activity and health monitoring (e.g., number of steps, heart rate, sleeping pattern; 64.4%, n = 201), position tracking via GPS signal (66.7%, n = 208), the archiving of life events (i.e., collecting pictures, documents, or other data; 55.8%, n = 174), the tracking of consumption/health behavior (e.g., loyalty programs; 48.4%, n = 151), and performance measurements at work (9.3%, n = 29).

### Data analysis

We perform descriptive analyses of the perceived benefits and barriers as well as TAM-related acceptance criteria and privacy issues related to the use of ambient sensor-based lifelogging technologies using means (M) and standard deviations (SD); standard errors are provided in the graphs. Checking the internal consistency of the scales by means of Cronbach's Alpha ($\alpha$ >.7) ensured a satisfying quality of the measured constructs (i.e., technology acceptance according to TAM [18], general privacy perceptions). To find out how privacy is predicting the technology acceptance, we calculated stepwise regression analyses. We set the level of statistical significance (p) at the conventional level of 5%.

### Results

In the following, we present participants' attitudes towards using sensor-based technology in home environments in terms of technology acceptance and their perceptions of personal privacy. In the first part, we display the evaluations of perceived benefits and barriers, and we analyze the sensor-related technology acceptance in line with TAM [18]. In the second part of the

Results section, we depict considerations as to privacy and its role for the use of ambient sensor-based technologies and applications. Within the third part, we estimate the relationships between the main research variables of this study.

### Evaluations of user acceptance

**Perceived benefits and barriers.** Thinking of benefits when using different sensors at home, participants especially appreciate the function of possible notifications in case of emergency. The resulting means (Fig 4) indicate that items describing the emergency reach the highest values on the scale (>5 of 6 possible points; responses < 3.5 were considered disagreement and values > 3.5 were considered agreement). The individual sense of security (M = 4.55, SD = 1.18) and safety feeling for the relatives (M = 4.78, SD = 1.17) are also clearly appreciated. In addition, the ability to live independently (M = 4.57, SD = 1.18) and an untroubled use of the technology at home (M = 4.44, SD = 1.27) are perceived as beneficial, contributing to an accepted use of ambient sensors in home environments.

Looking at findings for perceived barriers (Fig 5), it is noticeable that little agreement resulted for the studied items (all means are below the value of 3.5). According to the analysis, participants felt mostly uncertain with respect to the false alarms (M = 3.48, SD = 1.31), but they would use the technology not only in the event of a health-related restriction (M = 2.68, SD = 1.39). The respondents' positive attitude towards the sensors is also confirmed by the still quite low means resulting in the assessment of being monitored (M = 3.12, SD = 1.52) and regarding the potential fears of too high costs (M = 3.14, SD = 1.31). Participants disagreed that they would be disturbed by sensors installed in different rooms (M = 3.32, SD = 1.45) and their privacy concerns were also limited at this point (M = 3.31, SD = 1.45), although these aspects could potentially have a negative impact on the respondents' acceptance.

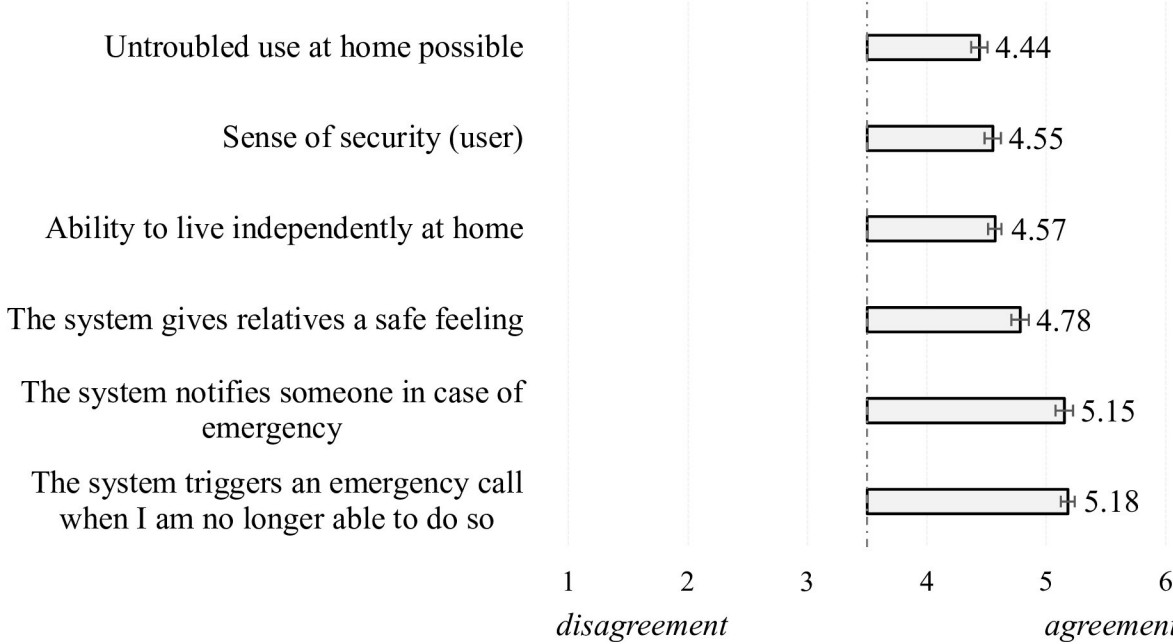

## Benefits of using sensor-based technology

Fig 4. **Perceptions of benefits when using sensor-based technology.**

## Barriers of using sensor-based technology

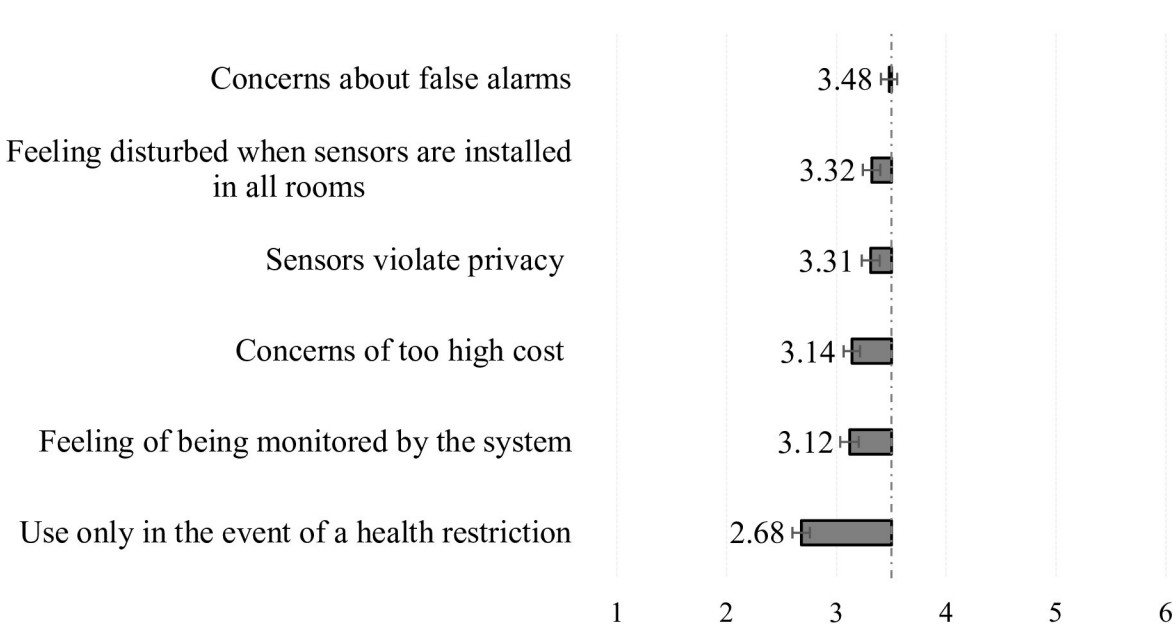

**Fig 5. Perceptions of barriers when using sensor-based technology.**

The positive perceptions of benefits and at the same time disagreements to possible barriers present a coherent picture, mirroring overall positive attitudes towards the ambient use of sensor-based technology.

**Technology acceptance according to TAM.** In order to underpin the aspect of acceptance in a more theoretically sound manner, the crucial technology acceptance criteria according to TAM [18] were examined.

The resulting means are depicted in Fig 6: While the mean values for the perceived ease of use (M = 8.75, SD = 1.97) and the ease of use (M = 8.79, SD = 2) clearly lie within the range of agreement (values >6.5), the intention to use ambient sensor-based lifelogging is rather restrained (M = 6.55, SD = 2.68), yet not rejected. In concrete terms, this means that the (potential) users perceive the ambient sensors as useful and predominantly easy to use, but they represent a rather neutral attitude towards the actual use of this technology so far.

### Perceptions of privacy

In the next step of the statistical analysis, we consider the aspect of privacy and its role for the use of ambient sensor-based applications. For this purpose, we firstly asked the survey participants about the importance of personal privacy in using sensor-based lifelogging in general as well as in terms of individual activities in different rooms at home. Secondly, we analyzed which role plays health condition in the perception of privacy. And finally, we reveal in which relation perceptions of privacy are connected with the technology acceptance of the assistive technology.

**Privacy: A curse or a blessing?.** In our questionnaire, participants were asked to assess their privacy related to the use of sensor-based technology in their private homes. The majority of the respondents agreed that privacy protection should have the first priority (M = 4.39,

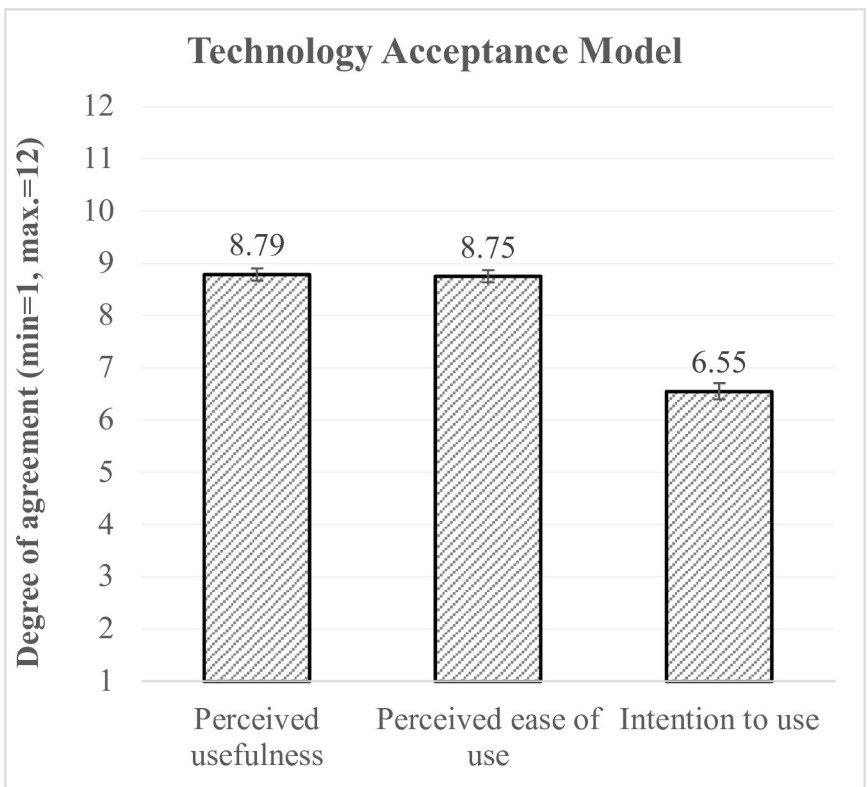

**Fig 6. Evaluations of the technology acceptance criteria according to TAM [18].**

SD = 1.4) and they also agreed that they would use sensor-based lifelogging only as long as their privacy is protected (M = 4.38, SD = 1.4). As it is shown in Fig 7, participants were overall relatively neutral towards recording for their own purposes (M = 3.5, SD = 1.5).

But which activities, or which areas in the private environments, cause the reluctance by users regarding the addressed technology? According to the results presented in Fig 8, the use of sensor-based technology in the home environment was truly critical only in two rooms, which insinuate a high intimacy requirement anyway. These rooms are the bathroom (M = 76.4; SD = 27.9, where 100 is the most critical value) and the bedroom (M = 76.3; SD = 26.9) showing almost identical means resulting from the assessments of the possible recordings of personal activities in these rooms. As less critical respondents perceived the stay in the kitchen (M = 38.1; SD = 29.3) as well as duration (M = 44.4; SD = 30.7) and frequency of stays (M = 43.3; SD = 33.1), and the timing of entering and leaving different rooms (M = 45.4; SD = 33.1). The other examined activities were perceived as rather neutral.

**What role does privacy still play in the use of the technology when it comes to health?.** Or maybe we should just ask the other way around: Which role does health condition play in the perception of privacy? Individual health was found to play a decisive role in the acceptance and use of health information technologies (e.g., [41, 42]). For this reason, the current study additionally examined how participants assess the relevance, and thus possibly the disclosure, of their health in relation to the privacy issues resulting from the logging of sensor-based data. To avoid biased evaluations we randomly used statements, which are both privacy-affirming and privacy-denying in the presence of a medical condition or physical limitation; for details see Fig 9.

In the comparison of how important privacy is in relation to one's own health, the outcomes reveal that health clearly takes the first position. Survey respondents decisively rejected

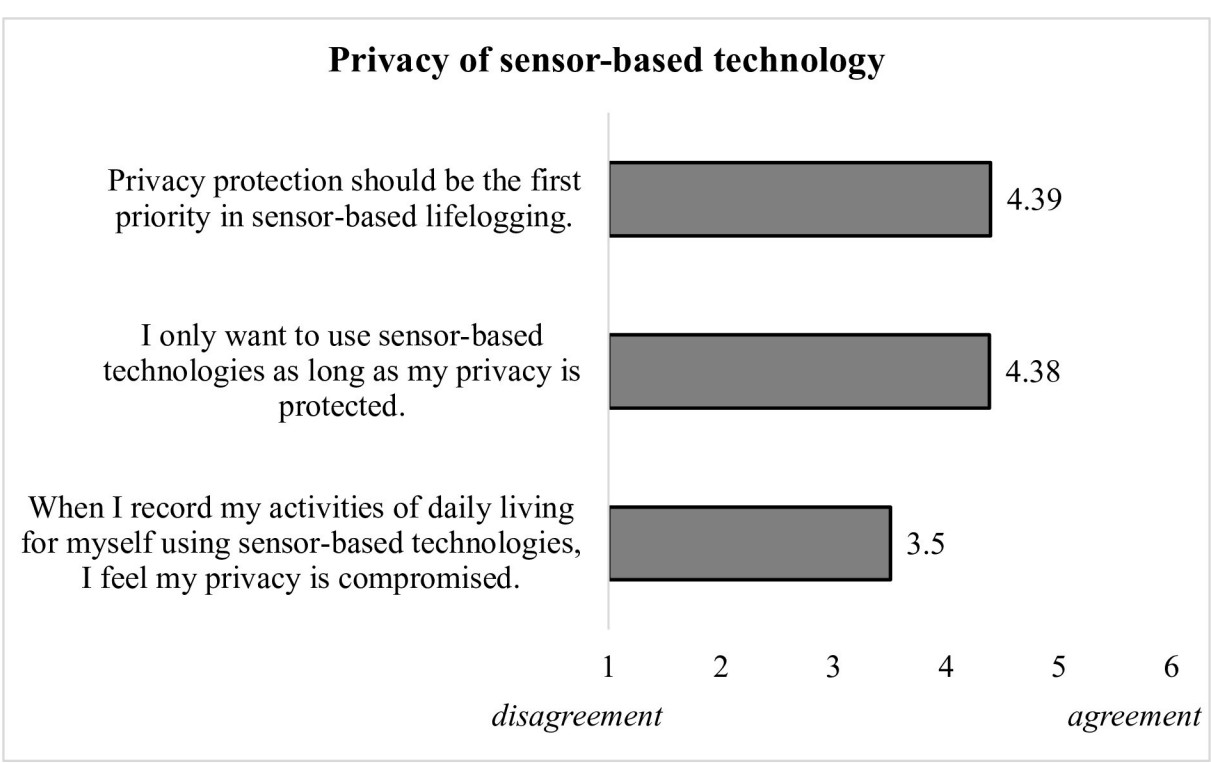

**Fig 7. Evaluations of privacy in the context of sensor-based technology.**

the higher importance of privacy (M = 2.5, SD = 1.2) and their persistence as to privacy protection in the light of a health emergency (M = 2.5, SD = 1.2). In contrast, according to the results, they would disclose the necessary information to important others when this would prevent an illness (M = 4.7, SD = 1.1), and they would make their data transparent in the situation of acute danger (M = 4.4, SD = 1.3).

### Relations between technology acceptance and the perceptions of privacy

For the final step of the statistical analyses, the question remains to what extent perceptions of privacy determine the acceptance of the use of sensor-based technology in home environments. For this purpose, we used the previously described construct 'privacy and the health data' (see Methods section) in connection with the key criteria of technology acceptance model.

A stepwise regression analysis revealed that privacy is significantly related to the accepted use of the technology. The model was statistically significant, $F(1,299) = 104.52$, $p \leq .001$, and accounted for approximately 26 percent of the variance of acceptance ($R^2 = .260$, adjusted $R^2 = .257$). The privacy as a predictor received the weight of $\beta = .51$ ($t(299) = 10.22$, $p \leq .001$), making a strong unique contribution to explain the technology acceptance of ambient sensors.

As can be seen in Table 2, privacy explained the variance of the perceived usefulness for the most (25.4%), followed by the intention to use (15.6%) and the perceived ease of use (7.8%), thus making its role as a significant predictor for the acceptance evident.

### Discussion

Smart technologies are becoming an increasingly important part of our lives. Miniaturized computer technology makes objects, in which it finds its way, part of a connected information

### Wariness about privacy in recording various activities

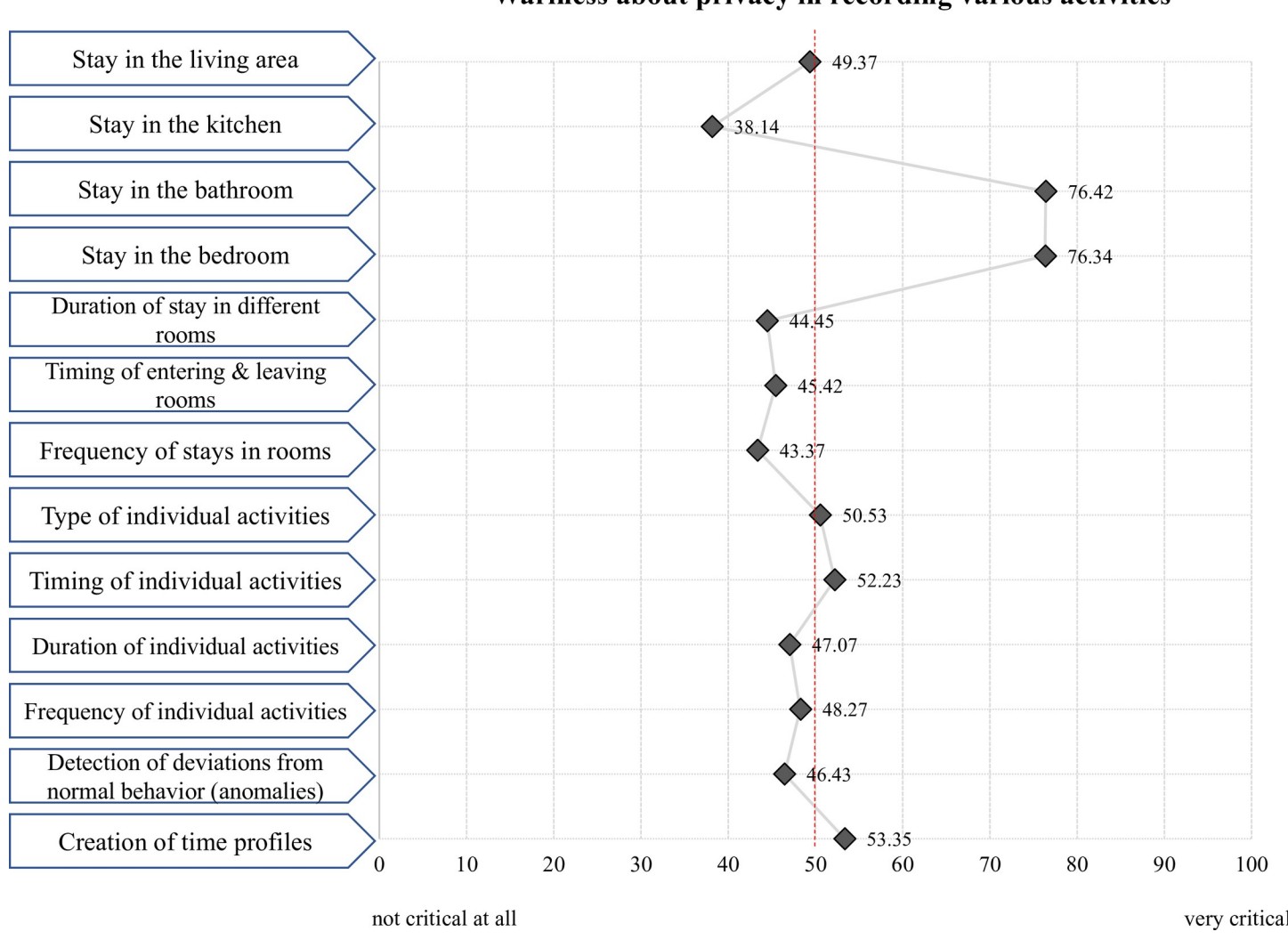

not critical at all                                                                          very critical

**Fig 8. Wariness about privacy issues when using sensor-based systems.**

and communication system and has the potential to continuously log personal data [43]. Sensor-based technologies that are not only available in the form of wearable devices but can also be implemented in the environment and be networked via mobile communications, enable to collect data from the surroundings without the user's active intervention and can exchange data, discreetly supporting the users in many areas of life.

The potentially life-enriching benefits of sensor-based lifelogging technologies, as described in the presented study, can be useful for the aging society and can improve autonomy as well as medical treatment and care of their users. Nevertheless, feasibility alone does not guarantee success in the implementation and sustainable use of such assistive lifelogging applications. To fully exploit the potential offered by these technologies and to ensure a long-term implementation, it is thus necessary to investigate possible acceptance issues from the perspective of (potential) users. To gain insights into the significant influences on the adoption of sensor-based lifelogging, we empirically investigated concrete acceptance factors as well as privacy issues—previously identified as critical in the lifelogging context—and the interrelationship between these variables. In the following, we discuss the findings and outline the need for

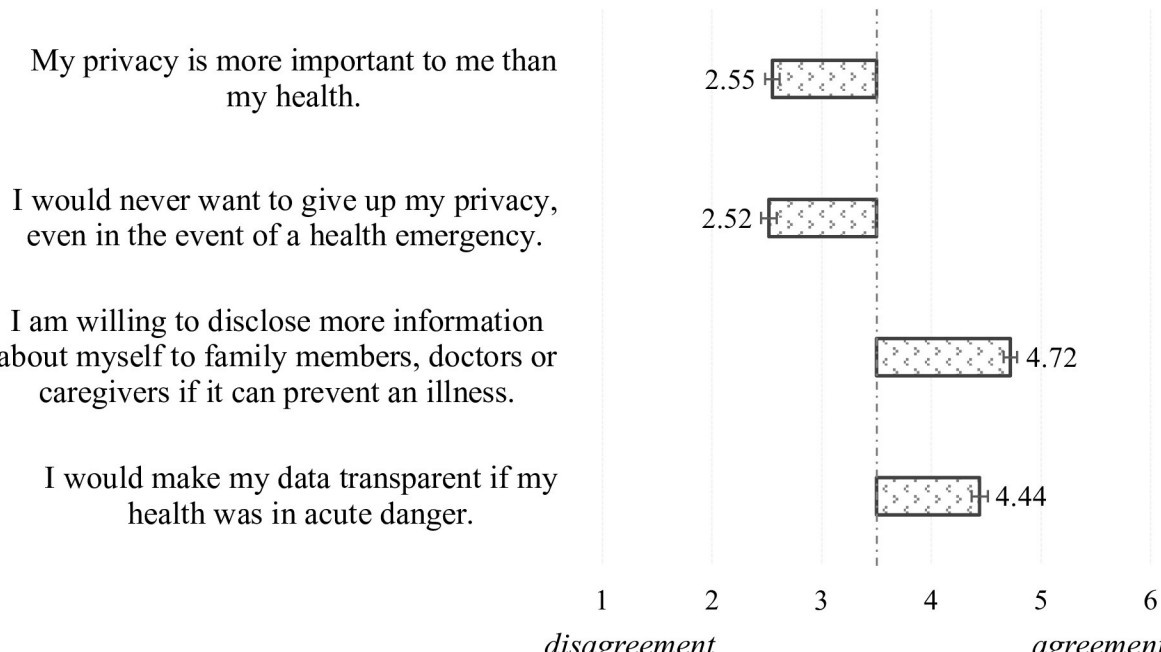

**Fig 9. The relevance of health in the perception of privacy when using sensor-based technology.**

further research in this context, considering at the same time the limitations of the presented study.

### User-centered preferences and acceptance-relevant considerations

According to our results, German adults eminently appreciate the benefits of sensor-based life-logging: In the ranking, the highest approval is assigned to the benefit of emergency functions, but the sense of safety and the ability of an independent living in the own four walls, that are ensured by the use of the technology, are also highly appreciated. These outcomes resonate with previous research in assistive lifelogging (e.g., [26, 29, 31]). On the side of the perceived barriers when using sensor-based applications, the participants mostly disagreed with the selected items. The highest uncertainty is noticeable with respect to the possible false alarms; invasion of privacy as well as the saturation of sensors in all living spaces could also pose potential drawbacks for the acceptance. However, from the perception of benefits and barriers, opinions about the assistive lifelogging technologies in private home environments are basically positive.

**Table 2. Results of a regression analysis for the acceptance of sensor-based technology in home environments ($N$ = 312; $^{**}p \leq .001$; VIF = variance inflation factor $< 10$).**

| | Predictor | Adj. $R^2$ | β | t | VIF | ANOVA |
|---|---|---|---|---|---|---|
| Acceptance of sensor-based technology | Privacy and/vs. health | 25.7% | .51 | 10.22** | 1.0 | $F(1,299) = 104.52, p \leq .001$ |
| *Perceived usefulness* | | 25.4% | .51 | 10.11** | 1.0 | $F(1,297) = 102.22, p \leq .001$ |
| *Intention to use* | | 15.6% | .40 | 7.40** | 1.0 | $F(1,291) = 54.73, p \leq .001$ |
| *Perceived ease of use* | | 7.8% | .28 | 5.10** | 1.0 | $F(1,295) = 25.97, p \leq .001$ |

To theoretically substantiate the aspect of technology acceptance, in our study we additionally examined the key acceptance criteria according to the renowned Technology Acceptance Model [18]. The survey results revealed that the potential users perceive the sensor-based lifelogging as useful and easy to use, nevertheless they intend to use it only hesitantly. This reluctance of using the ambient sensors may be due to the fact that these applications are not yet well-known and that their functionalities (e.g., drawing conclusions about daily habits and activities, identifying events that deviate from the daily pattern) are not immediately obvious to the respondents. On the other hand, assuming that the respondents were familiar with the benefits provided by the sensor-based lifelogging, as the sample was on average at just under 33 years still quite young, they may not have realized the added value for themselves—at least not at the current time in their lives. Thus, answering to the previously defined research questions (RQ1, RQ2) on the basis of our findings, we can state that the general acceptance of sensor-based technology is high. However, although the potential users seem to be guided more by the perceptions of benefits than drawbacks brought by the technology, and they clearly perceive its usefulness, the willingness to an immediate utilization is rather reserved.

In the next step, we focused on results concerning personal privacy, since in previous research the loss of privacy was identified as an important barrier towards the adoption of technologies in the AAL context (e.g., [39, 40]). Our findings revealed a high need for privacy protection: The majority of our sample assigned the first priority to the protection of their privacy when using sensor-based lifelogging and they agreed on using that technology only if their privacy is properly protected. When asked about specific rooms and types of sensor-logged activities, however, it turned out that only recordings in the bathroom and bedroom are perceived as critical, while other rooms are judged to be quite neutral for the lifelog. Nonetheless, the attitude about the highest priority for personal privacy changes when health issues come into play. Our survey respondents decisively rejected the higher importance of privacy in case of a health emergency: They would rather provide information needed to prevent an illness and disclose their health data in the situation of acute danger. These findings show that the perceptions of privacy are clearly situation- and context-dependent, playing an important role in the adoption of sensor-based ambient technologies (RQ3).

Furthermore, as for the acceptance of such technology in home environments we found, still considering the aspect of health, that privacy protection accounts to 26% for the willingness to use, and thus for an accepted adoption of, assistive sensor-based lifelogging technologies. This result of our study indicates that besides other factors, presumably having a decisive impact on acceptance, privacy protection significantly influences the adoption of such assistive technologies (RQ4).

With this in mind, we can state that potential users generally show an accepting attitude towards assistive sensor-based logging in their home environments and require high privacy protection standards for their personal data. Nevertheless, there is also the willingness to lower these requirements in the event of health-related changes in life situations.

## Limitations and future work

In addition to the relevant findings that this study has revealed in the context of a specific technology and use, some limitations and, in particular, the need for future research should be also addressed.

From the methodological point of view, an important limitation of the results pertains to the relatively young sample. Since young users predominantly have been shown to have greater technical expertise [44, 45] and perceived familiarity with new technology [46], this is probably also true, at least to some extent, for the investigated sensor-based lifelogging. In future studies,

therefore, the opinions regarding acceptance and privacy requirements are to be explored primarily from the perspective of older users, as they are the main target user group for this kind of assistive technology. Here, it is even conceivable that there are significant differences between healthy aging and chronically ill elders, which should be deliberately taken into account in the studies.

There are also other arguments in favor of focusing on older users in the described research context: Resulting from our findings, the willingness to an immediate use of sensor-based technology is rather reserved. At this point, we can only hypothesize the reasons for such an attitude, especially because of the resulting high general acceptance. Future research has thus to find out which factors or circumstances caused this finding. Furthermore, the question emerges whether this is a generally valid outcome or it is due to the specific characteristics of the participating respondents. It is conceivable that information derived from the everyday life of older persons or those in need of care and the conclusions drawn from the technology about their habits and activities will be perceived as very beneficial in the group of older users. Thus, future work should evaluate opinions of predominantly older users, who are confronted with a necessity of assistance in some areas of their life.

## Conclusions

The results of our empirical study demonstrate that the participants as potential users generally represent an accepting mindset towards ambient sensors that log different parameters in their living environments. The advantages and the perceived usefulness of the technology strongly outweigh the possible weaknesses. However, we must also note that the intention to use sensor-based lifelogging technologies is still relatively low. Technology acceptance is furthermore determined—to a non-trivial extent—by privacy protection, and users require a high privacy protection by default; even though they are willing to accept more transparency of personal data in the event that they suffer a health decline.

## Supporting information

**S1 Data.**
(CSV)

## Acknowledgments

We would like to thank all participants for sharing their opinions on using sensor-based technologies in home environments.

## Author Contributions

**Conceptualization:** Wiktoria Wilkowska, Julia Offermann.

**Data curation:** Wiktoria Wilkowska, Julia Offermann.

**Formal analysis:** Wiktoria Wilkowska.

**Funding acquisition:** Susanna Spinsante, Martina Ziefle.

**Investigation:** Wiktoria Wilkowska, Julia Offermann.

**Methodology:** Wiktoria Wilkowska, Julia Offermann.

**Project administration:** Susanna Spinsante, Martina Ziefle.

**Resources:** Wiktoria Wilkowska, Julia Offermann.

**Software:** Wiktoria Wilkowska, Julia Offermann.

**Supervision:** Susanna Spinsante.

**Validation:** Wiktoria Wilkowska, Julia Offermann.

**Visualization:** Wiktoria Wilkowska.

**Writing – original draft:** Wiktoria Wilkowska, Julia Offermann, Susanna Spinsante, Angelica Poli.

**Writing – review & editing:** Wiktoria Wilkowska, Julia Offermann, Susanna Spinsante, Angelica Poli.

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
