## [Decision Letter · Decision Letter 0]

1 May 2022

PONE-D-21-35533Analyzing Technology Acceptance and Perception of Privacy in Ambient Assisted Living for Using Sensor-Based TechnologiesPLOS ONE

Dear Dr. Wilkowska,

Thank you for submitting your manuscript to PLOS ONE. After careful consideration, we feel that it has merit but does not fully meet PLOS ONE’s publication criteria as it currently stands. Therefore, we invite you to submit a revised version of the manuscript that addresses the points raised during the review process. Please submit your revised manuscript by Jun 15 2022 11:59PM. If you will need more time than this to complete your revisions, please reply to this message or contact the journal office at plosone@plos.org. Please include the following items when submitting your revised manuscript:A rebuttal letter that responds to each point raised by the academic editor and reviewer(s). You should upload this letter as a separate file labeled 'Response to Reviewers'.A marked-up copy of your manuscript that highlights changes made to the original version. You should upload this as a separate file labeled 'Revised Manuscript with Track Changes'.An unmarked version of your revised paper without tracked changes. You should upload this as a separate file labeled 'Manuscript'.If applicable, we recommend that you deposit your laboratory protocols in protocols.io to enhance the reproducibility of your results. Protocols.io assigns your protocol its own identifier (DOI) so that it can be cited independently in the future. For instructions see: https://journals.plos.org/plosone/s/submission-guidelines#loc-laboratory-protocols. Additionally, PLOS ONE offers an option for publishing peer-reviewed Lab Protocol articles, which describe protocols hosted on protocols.io. Read more information on sharing protocols at https://plos.org/protocols?utm_medium=editorial-email&utm_source=authorletters&utm_campaign=protocols.

We look forward to receiving your revised manuscript.

Kind regards,

Giovanni Ottoboni, Psy, PhD

Academic Editor

PLOS ONE

Journal Requirements:

This work is part of the PAAL-project (Privacy-Aware and Acceptable Lifelogging services for older and frail people). In particular, the support of the Joint Programme Initiative More Years, Better Lives (award number: PAAL_JTC2017), the German Federal Ministry of Education and Research (grant no: 16SV7955), and the Italian Ministero dell’Universitá e della Ricerca, (CUP: I36G17000380001) is gratefully acknowledged.

Reviewers' comments:

Reviewer's Responses to Questions

**Comments to the Author**

1. Is the manuscript technically sound, and do the data support the conclusions?

Reviewer #1: Yes

Reviewer #2: Yes

2. Has the statistical analysis been performed appropriately and rigorously? 

Reviewer #1: Yes

Reviewer #2: Yes

3. Have the authors made all data underlying the findings in their manuscript fully available?

Reviewer #1: No

Reviewer #2: Yes

4. Is the manuscript presented in an intelligible fashion and written in standard English?

Reviewer #1: Yes

Reviewer #2: Yes

5. Review Comments to the Author

Reviewer #1: The paper describes an interesting and fascinating aspect of the adoption of wearable device – the lifelogging - to support mobility and autonomy of people with disabilities. The exploratory study was focus on general population to investigated the users’ technology acceptance and the privacy perception. Results of the 312 participants showed that the privacy is considered the main aspect that involve the acceptance of the assistive technology.

P7 line 163: I invite the authors to describe the definitions of PEU and PU.

It is not clear how the authors present the devices to participants to be evaluated. Do they use at home as trial? Do they receive a short presentation? I invite the authors to add these information.

Reviewer #2: The manuscript reports interesting aspects on technology acceptance and privacy perceptions in private environment of German adults. Moderate revisions are necessary to improve the value of the manuscript. The abstract has to report the name of the survey and technology adopted briefly. A more concise Introduction should implement the reading from the Journal audience. It is too long and you may not focus on the main topics. Some typos are also present: please see for instance line 257, May instead of “Mai”. Check carefully the whole ms for that. Finally, I suggest to summarize survey parts into a Table in order to allow readers a fast knowledge and reduce the length of the report.

6. PLOS authors have the option to publish the peer review history of their article (what does this mean?). If published, this will include your full peer review and any attached files.

Reviewer #1: No

Reviewer #2: No

---

## [Author Response · Author response to Decision Letter 0]

18 May 2022

Dear Editor, dear Reviewers,

Thank you very much for your valuable comments and the opportunity to revise our paper entitled “Analyzing technology acceptance and perception of privacy in ambient assisted living for using sensor-based technologies”. We are very thankful for your and the reviewers’ suggestions which lead to significant improvements of our manuscript. We addressed all comments in our best possible way and provided our responses within a table (for a better overview) indicating in which parts of the manuscript the corresponding changes are located. Please, find our responses below.

Reviewer 1:

“The paper describes an interesting and fascinating aspect of the adoption of wearable device – the lifelogging - to support mobility and autonomy of people with disabilities. The exploratory study was focus on general population to investigated the users’ technology acceptance and the privacy perception. Results of the 312 participants showed that the privacy is considered the main aspect that involve the acceptance of the assistive technology.”

Response: Thank you very much for your review and your valuable suggestions. We have included accordingly the particular modifications into our manuscript as reported below.

“P7 line 163: I invite the authors to describe the definitions of PEU and PU.“

Response: Thank you for this great suggestion. We have now added the definitions as suggested. (Location: Technology acceptance of sensor-based lifelogging technologies, Lines 160–165)

“It is not clear how the authors present the devices to participants to be evaluated. Do they use at home as trial? Do they receive a short presentation? I invite the authors to add these information.” 

Response: Thank you for this comment. We actually have explicitly described how our participants were confronted with the concrete technology (lines 270–281), but for a better understanding we still supplemented information in the text and we now added pictures of the sensors that we presented to the participants embedded in a schematic structure overview of the online survey (Fig 2). (Location: Data collection, Lines 270ff and 299–300)

Reviewer 2:

"The manuscript reports interesting aspects on technology acceptance and privacy perceptions in private environment of German adults. Moderate revisions are necessary to improve the value of the manuscript.”

Response: We are thankful for your review and your valuable advices. We now included some modifications and hope that our work can soon publicly contribute to a more sustainable research and user-centered development of lifelogging technologies. 

“The abstract has to report the name of the survey and technology adopted briefly.”

Response: Thank you for this great suggestion. We complimented the missing information and we additionally paraphrased the abstract so that it is now clearer for the reader. (Location: Abstrakt)

“A more concise Introduction should implement the reading from the Journal audience. It is too long and you may not focus on the main topics.” 

Response: Thanks for the recommendation. We now shortened the Introduction section leaving there only the main topics and the logical consequences for the users of the investigated technology. (Location: Introduction)

“Some typos are also present: please see for instance line 257, May instead of “Mai”. Check carefully the whole ms for that.”

Response: We reviewed the whole manuscript more precisely for typo and syntax errors. (Location: Whole manuscript,

References)

“Finally, I suggest to summarize survey parts into a Table in order to allow readers a fast knowledge and reduce the length of the report.”

Response: Thank you for this suggestion. In accordance with the another reviewer’s comment, we now added a Figure (Fig 2), which presents a schematic structure of the online survey. While this has—unfortunately—not reduced the length of the manuscript at this point, it has substantially improved the comprehensibility and overview of our method. (Location: Method, Data collection, Lines 299–300)

Editor /Journal requirements:

“Please state what role the funders took in the study.”

Response: Thank you for this advice. We added your suggested statement “the funders had no role in study design, data collection and analysis, decision to publish, or preparation of the manuscript.” to the financial disclosure. (Location: Funding Statement)

“In your Data Availability statement, you have not specified where the minimal data set underlying the results described in your manuscript can be found. PLOS defines a study's minimal data set as the underlying data used to reach the conclusions drawn in the manuscript and any additional data required to replicate the reported study findings in their entirety. All PLOS journals require that the minimal data set be made fully available”

Response: Thank you very much for this comment and information. We prepared a data sheet referring to the results in our manuscript and uploaded it as “supporting information". (Location: Data Availability Statement)

“Please include your full ethics statement in the ‘Methods’ section of your manuscript file. In your statement, please include the full name of the IRB or ethics committee who approved or waived your study, as well as whether or not you obtained informed written or verbal consent. If consent was waived for your study, please include this information in your statement as well.”

Response: Thank you for this valuable advice. We added a full ethic statement at the beginning of the methods section. (Location: Methods)

“Please review your reference list to ensure that it is complete and correct. If you have cited papers that have been retracted, please include the rationale for doing so in the manuscript text, or remove these references and replace them with relevant current references. Any changes to the reference list should be mentioned in the rebuttal letter that accompanies your revised manuscript. If you need to cite a retracted article, indicate the article’s retracted status in the References list and also include a citation and full reference for the retraction notice.”

Response: Thank you for this information. In line with a thorough proofreading, we reviewed our reference list and confirm the references to be complete and correct.(Location: References)

---

## [Editor Report · Decision Letter 1]

25 May 2022

Analyzing technology acceptance and perception of privacy in ambient assisted living for using sensor-based technologies

PONE-D-21-35533R1

Dear Dr. Wilkowska,

We’re pleased to inform you that your manuscript has been judged scientifically suitable for publication and will be formally accepted for publication once it meets all outstanding technical requirements.

Kind regards,

Giovanni Ottoboni, Psy, PhD

Academic Editor

PLOS ONE
---

## [Editor Report · Acceptance letter]

23 Jun 2022

PONE-D-21-35533R1 

Analyzing technology acceptance and perception of privacy in ambient assisted living for using sensor-based technologies 

Dear Dr. Wilkowska:

I'm pleased to inform you that your manuscript has been deemed suitable for publication in PLOS ONE. Congratulations! Your manuscript is now with our production department. 

Kind regards, 

on behalf of

Professor Giovanni Ottoboni 

Academic Editor

PLOS ONE